# Effects of Terahertz Radiation on the Aggregation of Alzheimer’s Aβ42 Peptide

**DOI:** 10.3390/ijms24055039

**Published:** 2023-03-06

**Authors:** Lei Wang, Yuanyuan Cheng, Wenxia Wang, Jinwu Zhao, Yinsong Wang, Xumei Zhang, Meng Wang, Tianhe Shan, Mingxia He

**Affiliations:** 1The Center for Terahertz Waves, School of Precision Instrument and Opto-Electronics Engineering, Tianjin University, Tianjin 300072, China; 2State Key Laboratory of Precision Measuring Technology and Instruments, Tianjin University, Tianjin 300072, China; 3School of Pharmacy, Tianjin Medical University, Tianjin 300203, China; 4School of Public Health, Tianjin Medical University, Tianjin 300203, China

**Keywords:** THz radiation, Aβ, aggregation, molecular dynamics simulation, secondary structure

## Abstract

The pathophysiology of Alzheimer’s disease is thought to be directly linked to the abnormal aggregation of β-amyloid (Aβ) in the nervous system as a common neurodegenerative disease. Consequently, researchers in many areas are actively looking for factors that affect Aβ aggregation. Numerous investigations have demonstrated that, in addition to chemical induction of Aβ aggregation, electromagnetic radiation may also affect Aβ aggregation. Terahertz waves are an emerging form of non-ionizing radiation that has the potential to affect the secondary bonding networks of biological systems, which in turn could affect the course of biochemical reactions by altering the conformation of biological macromolecules. As the primary radiation target in this investigation, the in vitro modeled Aβ42 aggregation system was examined using fluorescence spectrophotometry, supplemented by cellular simulations and transmission electron microscopy, to see how it responded to 3.1 THz radiation in various aggregation phases. The results demonstrated that in the nucleation aggregation stage, 3.1 THz electromagnetic waves promote Aβ42 monomer aggregation and that this promoting effect gradually diminishes with the exacerbation of the degree of aggregation. However, by the stage of oligomer aggregation into the original fiber, 3.1 THz electromagnetic waves exhibited an inhibitory effect. This leads us to the conclusion that terahertz radiation has an impact on the stability of the Aβ42 secondary structure, which in turn affects how Aβ42 molecules are recognized during the aggregation process and causes a seemingly aberrant biochemical response. Molecular dynamics simulation was employed to support the theory based on the aforementioned experimental observations and inferences.

## 1. Introduction

Alzheimer’s disease (AD) is a common degenerative illness of the central nervous system, characterized by memory loss, an inability to care for oneself, etc. AD is becoming more common as the world population ages. According to the World Health Organization’s Global Status Report on Addressing Dementia in Public Health, there are currently approximately 55 million individuals living with AD. It is predicted that this number will increase to 78 million by 2030 and as many as 139 million by 2050. Families and societies bear a heavy burden from AD [1]. Numerous studies have been conducted on the causation and treatment of AD to reduce its occurrence and progression. It is widely thought that the accumulation of β-amyloid (Aβ) in the central nervous system is a key factor in the etiology of AD [2]. Aβ oligomer, which are the intermediate aggregated forms of the neurotoxic aggregates of Aβ, have high neurotoxicity, and the concentration of Aβ oligomer and the severity of cognitive deficits in AD patients are closely correlated [3,4,5]. Thus, understanding the variables that affect Aβ aggregation is essential for preventing and treating AD. A growing number of academics are currently studying the factors that affect Aβ aggregation, including the impact of electromagnetic radiation. The intensity of electromagnetic radiation that people are exposed to is increasing as a result of the widespread use of electromagnetic wave technology in a variety of industries. Some studies have shown that professionals who are exposed to high-intensity electromagnetic fields for extended periods of time are at high risk of developing AD [6,7,8], and an increasing number of researchers in various fields have also discovered that electromagnetic radiation can have a non-negligible negative impact [9,10,11]. It has been demonstrated that microwaves can affect several significant biological processes, including the pathogenesis of Alzheimer’s disease [12].

Terahertz waves are electromagnetic waves with frequencies between 0.1 and 10 THz and wavelengths between 3 μm and 3 mm. They are special electromagnetic bands with both electronic and optical properties that hold great promise for use in the fields of communication, defense, and medicine [13]. However, because terahertz technology is still in its infancy, its biosafety is not completely understood. The photon energy of terahertz waves is extremely low (1 THz ≈ 4.1 meV), far too low to destroy materials through ionization, yet it is exactly in the energy level range that corresponds to the secondary bonds that preserve the structural stability of biological macromolecules. Based on this specificity, it has been speculated that terahertz waves can interfere with an organism’s biochemical processes, thus posing an underestimated biosafety danger [14]. Numerous studies have shown that terahertz radiation affects the structure and operation of proteins. As indicated by the research by Govorun et al. [15], terahertz radiation has been shown to affect the stability of protein secondary structures since the 1990s. Researchers have recently investigated how enzyme activity changes in response to terahertz radiation. For instance, Homenko et al. [16] first discovered that the catalytic activity of alkaline phosphatase and the process of antigen-antibody binding can be adversely affected by terahertz waves with a frequency of 0.1 THz and a radiation intensity of 0.08 mW/cm^2^. Zhang et al. [17] have since demonstrated that exposure to 0.1 THz radiation for 20 min can decrease alkaline phosphatase activity.

Humans are extensively exposed to terahertz radiation, as terahertz technology is widely used. It is, therefore, extremely relevant to study the mechanism by which Aβ, a crucial protein connected to the pathogenesis of AD, aggregates in the presence of terahertz radiation. 

In this investigation, a quantum cascade laser with an output frequency of 3.1 THz was used as a source of terahertz radiation. β-amyloid 1-42 (Aβ42), which is more neurotoxic and prone to aggregation, was selected as a representative of the Aβ family for this study. The results showed that terahertz waves can significantly promote the aggregation of Aβ monomers. However, as the incubation time increased, the ability of terahertz waves to promote the aggregation of Aβ diminished and eventually changed to inhibition. We deduce that terahertz radiation may affect the aggregation activity of Aβ by affecting their spatial structure and consequently their aggregation activity because the secondary bonds of biological macromolecules and the energy levels of collective vibration and rotation are mostly located in the terahertz band, and numerous studies have shown that terahertz waves can affect the assembly process of biological macromolecules [18,19]. We used molecular dynamics simulations to examine the structural changes of Aβ monomers and oligomers under terahertz electric fields at the theoretical level and to explain the experimental observations to further understand the effect of terahertz radiation on proteins.

## 2. Results and Discussion

### 2.1. Effect of Terahertz Radiation on Various Aβ42 Forms

#### 2.1.1. Thioflavin-T Fluorescence Experiments

The protein was activated by two hours of incubation at 37 °C in an Aβ42 solution at a concentration of 100 μM. The irradiation time was set to 2, 4, and 6 h for each of the three radiation groups. After irradiation, the relaxation process was allowed to proceed for 2 h. ThT was added, and the fluorescence intensity was measured to observe the effect of terahertz radiation on Aβ42 during aggregation. The experiments were repeated three times for each group. For comparison, the fluorescence intensities of all control groups were normalized to obtain the final values. 

Figure 1a illustrates the considerable enhancing effect of terahertz wave radiation on Aβ42 aggregation, but the promoting effect of terahertz waves on the Aβ42 aggregation process gradually decreased with the increase in terahertz waves irradiation time. This is a very interesting phenomenon; logically speaking, longer radiation exposure equals a higher radiation dose, and increasing the terahertz radiation dose should tend to gradually amplify the associated biological effects. However, the experimental results were the exact opposite of what one might expect.

We then observed the aggregation properties of the Aβ42 monomer under terahertz radiation. We directly prepared 100 μM Aβ42 solutions and conducted radiation experiments without any pre-treatment. To ensure the dominance of the monomer, we irradiated for only 2 h, given the high rate of Aβ42 aggregation. The results are shown in Figure 1b.

We found that the aggregation process of the Aβ42 monomer was significantly promoted by terahertz wave radiation and to a greater extent than the groups shown in Figure 1a. This result implies that Aβ42’s facilitation of terahertz waves may be related to Aβ42 aggregation.

Next, we created a model for several aggregation states of Aβ42 based on various incubation times, and we irradiated Aβ42 in various aggregation states to confirm the relationship between the response of Aβ42 under terahertz radiation and the aggregation state. In the previous step, we made the Aβ42 solution with a concentration of 100 μM first, and after varying the incubation times, five incubation time groups of 0 h, 2 h, 8 h, 16 h, and 32 h were set, where the 0 h incubation group was used as a model of monomer aggregation, the 2 h incubation group as a model of oligomer aggregation, the 8 h and 16 h incubation groups as the model of oligomers forming protofibrils, and the 32 h incubation group as the end of aggregation. The aforementioned experimental groups were exposed to terahertz radiation for 2 h. The device then measured the differences in fluorescence intensity for each group, normalized the data for all groups using the control group as the reference value, and displayed the relative fluorescence intensities for all experimental groups, as shown in Figure 1c. 

Comparing the experimental results of various incubation experimental groups, we discovered that terahertz radiation significantly promoted the aggregation of monomers to oligomers (0 h group and 2 h group), and the degree of promotion exhibited a decreasing trend as the number of oligomers increased (i.e., as the incubation time increased). Terahertz radiation even had a significant inhibitory effect on the aggregation of Aβ42 into protofibrils (8 h group and 16 h group). After 32 h of incubation, Aβ42 aggregation reached its peak height, and the disparities between the fluorescence readings of the experimental and control groups disappeared. 

According to the aforementioned experimental results, terahertz radiation can significantly promote the aggregation of monomers into oligomers while inhibiting the aggregation of the oligomers into protofibrils. Terahertz radiation only affected the aggregation process of Aβ42 and did not result in a significant reversal effect for the complete aggregation process.

#### 2.1.2. Morphological Analysis of Aβ42 Aggregates by Electron Microscopy

The above results indicate that exposure to terahertz radiation results in different effects on the aggregation of Aβ42 molecules in different states. To visualize this difference, we have, therefore, used transmission electron microscopy to observe the morphology of Aβ42 molecules after exposure to terahertz radiation, and the results are shown in Figure 2.

After 2 h of terahertz radiation, samples of the freshly prepared Aβ42 solution and the oligomer solution created after 6 h of incubation of the monomer were imaged. Figure 2a depicts the Aβ42 monomers that were allowed to spontaneously aggregate in the absence of radiation for 2 h, which resulted in only a few oligomer clusters. Figure 2b depicts the Aβ42 monomers that were exposed to radiation for 2 h, resulting in the formation of more diverse oligomer clusters of various sizes compared to the control group. Figure 2c,d of the oligomer solutions show that the Aβ42 oligomers could spontaneously form a protofibril-like structure, whereas the morphological characteristics of the oligomer solutions exposed to radiation for 2 h were significantly less distinct than those of the control group, despite the fact that there was a tendency to aggregate into protofibrils.

### 2.2. Effect of Terahertz Radiation on Cytotoxicity Caused by Aβ42 Aggregation

#### 2.2.1. MTT Assay

In the previous section, we showed that terahertz radiation promotes the aggregation of Aβ42 monomers while having a somewhat inhibitory effect on further aggregation into oligomers. However, it is still unclear whether the accumulation of oligomers induced by terahertz radiation can result in appreciable cytotoxicity. Thus, to examine this, we selected the commonly used AD model cells and PC12 cells for modeling in order to approximate the physiological conditions.

First, to determine whether terahertz radiation exerted significant cytotoxicity toward PC12 cells, we evaluated the cell viability following radiation using the MTT method for cell samples without Aβ42 as a radiation-negative control. As shown in Figure 3a, PC12 cells did not exhibit any discernible cytotoxicity following exposure to terahertz radiation. 

After splitting the experimental group into monomer intervention and oligomer intervention groups, we partially modeled and tested the results using the experimental methodology; the outcomes are shown in Figure 3b,c. It can be seen that in the radiation experiment of monomer intervention, the cell viability of the radiation group was significantly lower than that of the experimental group, and the viability decreased by approximately 10%, indicating that terahertz radiation promoted the aggregation of Aβ42 monomers, and the aggregates can exert significant cytotoxicity; however, for oligomers, the radiation group and the control group did not exhibit significant differences in the cytotoxicity of the aggregates. The combined results of the experiments at the cellular and molecular levels revealed that the enhancing effect at the cellular level was weaker than that at the molecular level. This was because Aβ42 was used at a lower concentration in the cellular experiments and because living cells have a certain level of resistance, which results in a reduced response to toxic effects.

#### 2.2.2. Fluorescent Staining of Live Cells

Given that significant differences in cytotoxicity were observed following radiation with only Aβ42 monomer intervention in the MTT experiments, we further examined the post-irradiation cytotoxicity of the Aβ42 monomer by live-cell fluorescence staining with calcein. As shown in Figure 4, the number of PC12 cells was highest in the control group, i.e., in the absence of treatment, than in the group that received only Aβ42, whereas the number of cells in the irradiation group decreased drastically after being exposed to Aβ42 and terahertz radiation.

### 2.3. Mechanism of the Effect of Terahertz Radiation on the Aggregation of Aβ42

#### 2.3.1. Analysis of the Molecular Trajectories of Aβ42 under the Action of Terahertz Radiation

The changing pattern of Aβ42 aggregation under terahertz radiation exposure was discovered in Section 2; however, the aforementioned investigations were unable to shed any light on the underlying molecular process. To explore this further, we used molecular dynamics to examine the dynamic behavior of the Aβ42 monomer and pentamer molecule within 20 ns under the terahertz field, and we sought to determine how terahertz radiation influences aggregation behavior at a structural level. Figure 5 displays the outcomes of our initial analysis of the dynamical trajectories of monomers and oligomers separately in the presence or absence of terahertz electric field intervention. For the molecular models shown in Figure 5, the default coloring of the VMD software was used to distinguish the various amino acids with different colors.

As shown in Figure 5a, in the control group, the Aβ42 monomer tended to spontaneously form an α-helix structure to maintain the stability of its structure, whereas in the system with a terahertz electric field intervention, this process was inhibited, causing the secondary structure of Aβ42 monomers to tend toward disorder, which is detrimental to structural stability. As a result of the unstable structure of Aβ42, additional hydrophobic groups may become exposed, which would enhance the monomer aggregation process. In addition, we observed that the monomer exhibits a tendency to form a β-sheet structure, a configuration that is closely associated to aggregation but cannot persist steadily in the presence of a terahertz electric field. The simulated behavior of the hydrophobic fragments in the oligomeric form of Aβ42 within 20 ns is shown in Figure 5b. It can be seen that the oligomer could essentially maintain the stability of its β-sheet structure in the system without intervention, and the aggregation morphology did not significantly change within 20 ns, whereas in the system with terahertz electric field intervention, it was unable to maintain a stable secondary structure and tended to become disordered again. It is well known that the β-sheet structure is crucial for preserving the hydrophobicity of Aβ42 oligomer and that the terahertz electric field may quickly destroy the β-sheet structure of Aβ42 molecules within 5 ns. Thus, it would appear that this could be the cause of the terahertz radiation’s inhibitory influence on the oligomeric system’s tendency to aggregate. Additionally, we found that the terahertz electric field could disassemble the initially more stable sheet structure in the oligomer simulation, but it did not cause the aggregates to depolymerize. Similar conclusions were obtained experimentally. 

There are still some gaps in our research; for example, the concentration of the PBS utilized in the experiment and the physiological salt solution used in the simulation differ by a factor of ten, and the PBS contains more types of ions than the ionic environment used in the simulation. The varied ion concentrations may have an impact on the system’s polarization even if the salt ions used in the models and tests will not, in theory, interact with the proteins. Additionally, various salt ions will alter the pH of the solution, which may have an impact on the protein’s secondary structure. According to the aforementioned, the track might not accurately represent the genuine protein track in each system. However, because the ionic concentration is constant throughout all simulated systems, it is possible to conclude that the presence of an electric field will have an impact on the Aβ42’s structure.

The root mean square deviation (RMSD), which can be used to measure the stability of a protein by contrasting it with its initial structure, is frequently employed to reflect variations in protein structure during dynamic simulation [20]. The RMSDs of the Aβ42 monomer in the system with and without a terahertz electric field within 20 ns are depicted in Figure 6a and Figure 6b, respectively. The presence of a terahertz electric field makes the molecular structure of the Aβ42 monomer less stable and therefore more prone to aggregation, as can be seen in Figure 6, where the mean RMSD is significantly higher and more volatile in the presence of a terahertz electric field intervention compared to Aβ42 in the system without electric field intervention. Similar patterns can be seen in the RMSD curves of the Aβ42 oligomer in Figure 6c,d.

#### 2.3.2. Secondary Structure Analysis

As shown in Section 2.1.1, the secondary structures of the Aβ42 molecule was gradually disrupted by the terahertz electric field, and it is well known that conformation governs the biological function of biological macromolecules. Next, we computed the variations in the number of secondary structures during the simulation based on the DSSP algorithm [21] to quantitatively analyze the changes in the secondary structures of the Aβ42 molecule under the influence of a 3.1 THz electric field. The results are shown in Figure 7. Under the influence of a 3.1 THz electric field, the diversity of the secondary structures formed by the Aβ42 monomer rapidly declined, and the number of residues involved in the formation of the secondary structures was dramatically reduced. The terahertz electric field unfolded the monomer molecule by decreasing the α-helix structure and increasing the random coil. In addition, we can see more clearly that the monomer molecule exhibits a tendency to form β-sheets when a terahertz electric field is present, but these β-sheets quickly evaporate. The same results were obtained for the oligomer; in particular, the quantity of β-sheet structures, which is directly associated with aggregation, quickly vanished. This conclusion validates the outcomes found in Section 2.1.1 and is comparable to that of the trajectory analysis in Section 2.3.1.

In summary, our results show that the presence of a terahertz electric field can greatly weaken the stability of the Aβ42 molecule, causing its structure to trend toward chaos and disorder and thus making it less functional.

#### 2.3.3. Hydrogen Bond Analysis

According to the above analysis, the secondary structures of the Aβ42 molecule were disrupted by the 3.1 THz electric field, thus reducing the stability of the Aβ42 molecule. As a result, we conclude that disorder can result from the hydrophobic groups of unstable Aβ42 molecule becoming disordered owing to the disruption of the structural order caused by the terahertz radiation. We further explored the interaction between Aβ42 and water molecules to examine the change in hydrophobicity of Aβ42 under the influence of a 3.1 THz electric field because the hydrophobicity of Aβ42 is intimately related to its aggregation process. The results are shown in Figure 8. 

We tallied the percentage of significant secondary structures and the average number of hydrogen bonds from three simulations with various starting velocities. Among these, it should be highlighted that the system reaches a stable condition after 5 ns, according to the results shown by the RMSD. We chose the system after 10 ns to determine the average number of hydrogen bonds. Table 1 displays the data in summary, including the average for each group and their relative standard deviation (RSD). As can be seen, the results exhibit good repeatability.

#### 2.3.4. Solvent-Accessible Surface Area Analysis

We investigated another physical measure of hydrophobicity, the solvent-accessible surface area (SASA) [22], to study the changes in the hydrophobicity of the Aβ42 molecules when exposed to a 3.1 THz electric field. The solvent-accessible surface area is the surface area of a biomolecule accessible to the solvent, measured in Å^2^, and was proposed by Lee and Richards in 1971; hence, it is called the Lee–Richards molecular surface. For the representation of this physical quantity, we used the region with atomic charges in the −0.2~0.2 interval as the hydrophobic region, and the other parts were used as hydrophilic regions. The results of the analysis are shown in Figure 9. For the monomeric molecule, the area of the hydrophobic region increased noticeably, while the size of the hydrophilic region shrank, demonstrating that the hydrophobicity of the Aβ42 molecule was increased by the 3.1 THz electric field. 

Because the oligomeric molecule’s SASA changed following the same pattern, the molecule’s overall hydrophobicity also increased, but the rate of aggregation decreased as a result of the disruption of the secondary structures linked to aggregation.

## 3. Materials and Methods

### 3.1. Samples and Reagents

Aβ42 was purchased from GL Biochem (Shanghai, China). Phosphate buffer solution (PBS, 0.01 M, pH 7.4) and Dulbecco’s modified Eagle’s medium (DMEM) were obtained from Corning Inc. (New York, NY, USA). Fetal bovine serum (FBS) was purchased from NEWZERUM (Christchurch, New Zealand). The MTT assay kit was purchased from Aladdin (Shanghai, China). Thioflavin T (ThT) was purchased from Zancheng Technology Co. (Tianjin, China). Phosphotungstic acid was obtained from HEOWNS Technology Co. (Tianjin, China). The adrenal pheochromocytoma (PC12) cell line was kindly donated by Prof. Zhang Xumei of Tianjin Medical University. 

### 3.2. Radiation Method

A terahertz quantum cascade laser (THz-QCL, Institute of Semiconductors, Chinese Academy of Sciences, Beijing, China) with a 3.1 THz frequency and 0.27 mW power served as the terahertz generator. The experimental samples were inoculated in a transparent polystyrene 24-well cell culture plate (Corning Inc., New York, USA) that was placed in an opaque chamber with an internal temperature of 37 °C, and the exposed group and the control groups were only separated by a distance of one well during the experiment. Two off-axis parabolic mirrors (OAP) and a reflector were used in the optical route design. After geometric optics measurements, the diameter of the terahertz beam at the endpoint was set to 1.5 cm, exactly covering the exposed group’s sample hole. The humidity of the environment where the irradiation system was kept was maintained below 10% to prevent significant energy attenuation during the propagation of terahertz waves. Figure 10 provides a schematic outline of the complete irradiation system.

### 3.3. Thioflavin T Fluorescence Test 

Monomerization procedure: Aβ42 powder was dissolved in hexafluoropropylene alcohol (HFIP) after being ultrasonically stirred in an ice bath and centrifuged at 4 °C. The precipitate was freeze-dried for the purification of Aβ42 monomer, followed by storage at −20 °C (this part was entrusted to GL Biochem). 

Preparation of Aβ42 solution: In brief, monomer Aβ42 powder was dissolved in NaOH solution (20 mM) and diluted in PBS to a concentration of 100 μM. It was stored at −20 °C and kept for no longer than a week.

ThT fluorescence test: ThT powder was dissolved in PBS solution to a concentration of 25 μM. After irradiation, the ThT solution was diluted with Aβ42 solution at a 20:1 volumetric ratio, and the ThT fluorescence intensity was measured using an F-7000 FL spectrophotometer (Tokyo, Japan). The excitation wavelength was set at 440 nm, and the emission wavelength was set at 480 nm. The excitation and emission slit lengths were 5 and 10 nm, respectively. The fluorescence of the PBS solution was used as the background fluorescence value.

### 3.4. Transmission Electron Microscopy

Following treatment using the aforementioned processes, the Aβ42 monomer or oligomer solution was diluted 50 times before being placed onto a copper grid with a 200-mesh ultra-thin carbon film copper mesh and allowed to dry in a cool environment. The sample was then stained with a 1% phosphotungstic acid solution, which was then placed over it in the shape of a sphere and left for 10 s before the liquid was aspirated. The sample was observed using a Hitachi HT7700 transmission electron microscope (Tokyo, Japan) at a voltage of 80 kV and a scale of 300 nm.

### 3.5. Cell Culture and Toxicity Assay

PC12 cells were used in the cytotoxicity assay and cultured in DMEM with 10% FBS and 100 U/mL penicillin-streptomycin. Briefly, PC12 cells were seeded separately in 96-well plates onto glass coverslips at a density of 1 × 10^4^ cells/well and incubated overnight. The cells were then incubated with a 5 μM Aβ42 solution. For the Aβ42 monomer intervention system, radiation was initiated immediately after the addition of Aβ42, whereas cells without radiation were used as the control group. Considering the decrease in Aβ concentration, we extended the radiation time for all the experimental groups. After further incubation for 24 h, MTT solution was added for 4 h and the plate was incubated at 37 °C. The medium in each well was then replaced with DMSO and shaken at 300 rpm for 5 min. The optical density (OD) at 490 nm was measured using an enzyme marker, and the cell viability was calculated as follows: $$\mathrm{Cell}\;\mathrm{viability}\;\left(\%\right)=\frac{\mathrm{OD}\;(\mathrm{sample})\;-\;\mathrm{OD}\;(\mathrm{blank})}{\mathrm{OD}\;(\mathrm{control})\;-\;\mathrm{OD}\;(\mathrm{blank})}\;\times \;100\%$$

### 3.6. Live-Cell Fluorescence Experiments

The PC12 culture conditions were the same as for 24-well plates, and 5 × 10^4^ cells were added to each well. Radiation was applied to the Aβ42 modeling group or control group, as mentioned in Section 3.5. Irradiation followed by 20 hours of culture, the culture solution was carefully removed, and calcein solution with a concentration of 0.5 M was added to each well, followed by co-incubation for 30 min. Live cells were observed under a Nikon Ti-U inverted fluorescence microscope (Tokyo, Japan).

### 3.7. Molecular Dynamics Simulations

The Protein Data Bank (PDB) database provided the structures of the Aβ42 monomer and oligomer, with database numbers 6szf [23] and 2beg [24] for monomer and oligomer, respectively, both of which selected the first frame of NMR as the initial structure, as shown in Figure 11. GROMACS software [25,26,27,28,29,30] was used to generate the topology files directly; GROMOS96 54a7 force field [31] was selected for the simulations of the Aβ42 molecules; and the SPC water model was used for both water models. Rectangular boxes with side lengths of 2 and 4 nm were then constructed for the monomer and oligomer, respectively, and filled with water molecules. Na^+^ and Cl^−^ concentrations of 0.15 M were added to the system to replace the equivalent number of water molecules to make the system electrically neutral and more representative of the physiological environment. Periodic boundary conditions in all directions were used.

According to the classical physical theory, the average molecule’s translational kinetic energy is given by the thermodynamic homogeneity theorem as $\frac{{\mathrm{mv}}^{2}}{2}=\frac{3}{2}k_{b}T$, where *T* is the temperature and *k_b_* is the Boltzmann constant. The electromagnetic wave can be divided into an electric field part and a magnetic field part, where the ratio of the electric field strength $\left|E\right|$ to the magnetic induction strength $\left|B\right|$ is the speed of light *c*, according to classical electromagnetic theory. In an electromagnetic field, the effects of an electric field and a magnetic field on a molecule are denoted by the expressions *q**E*** and *q**v** × **B***, where *q* and ***v*** stand for the charge and motion velocity of the molecule, respectively. Therefore, we calculated that the electric field’s force on the molecules is roughly 6 × 10^5^ times greater than the magnetic field’s, indicating that the electric field has a significantly greater impact on the molecule. Therefore, in this study, we solely take into account the impact of the electric field on the molecules and neglect the impact of the magnetic field [32]. Therefore, we used E(t)=A · u cos (ωt+φ) to describe the terahertz electric field, where ***u*** is the unit vector to describe the direction of the electric field, *A* is the electric field strength set to 2.5 V/nm, ω is the angular frequency (ω=2πν), and φ is the initial phase. 

A molecular dynamics simulation was performed once the model parameters were set. First, 500 steps of the simulation were performed to minimize the energy and eliminate unreasonable contact. Then, 100 ps of restrictive dynamics were applied to the protein to cause the water molecules to fully relax and reach the equilibrium state, which was utilized as the initial model for the lengthy dynamic simulation. For the long-time simulation, the dynamic calculation was still performed by the leap-frog method with a simulation time of 20 ns, and the step size was adjusted according to the calculated system. The Parrinello-Rahman pressure bath and velocity-rescale heat bath were used for the system; the SPME method was chosen for the electrostatic interaction calculation; and the van der Waals interaction was described by the cut-off method. The results were repeated more than three times using different initial velocities, including 0 and values chosen at random by the system. VMD software [33] was used to visualize the results of the molecular models and dynamic simulations, while Xmgrace software [34] was used to statistically construct data plots. 

### 3.8. Statistical Analysis

The data presented in this study were standardized for all control groups to facilitate comparison, and the means of the experimental group data are presented as relative values. The data were collected from at least three independent measurements conducted for each experimental condition. One-way ANOVA was used to analyze the results of data from more than two groups, whereas the *t*-test was used to analyze data from only two groups. The criterion for statistical significance was set at *p* < 0.05.

## 4. Conclusions

We examined how terahertz radiation affects the aggregation of Aβ42 peptides, which are linked to Alzheimer’s disease. By combining terahertz radiation with ThT fluorescence staining, transmission electron microscopy imaging, and cytotoxicity tests, we discovered that terahertz radiation with a frequency of 3.1 THz and a power density of about 0.18 mW/cm^2^ could significantly promote the nucleation aggregation process of Aβ42 monomers while also significantly inhibiting the fibrillation aggregation phase of Aβ42 oligomers.

We also employed a molecular dynamics strategy to examine the dynamic behavior of the Aβ42 monomer and the Aβ42 oligomer under the influence of a terahertz electric field. This included examining changes in dynamic trajectories, RMSD, secondary structures, the number of hydrogen bonds formed by Aβ42 and water molecules, and SASA. These results demonstrate that the conformation of Aβ42 is perturbed by a 3.1 THz electric field, which consequently affects its aggregation function. 

## Figures and Tables

**Figure 1 ijms-24-05039-f001:**
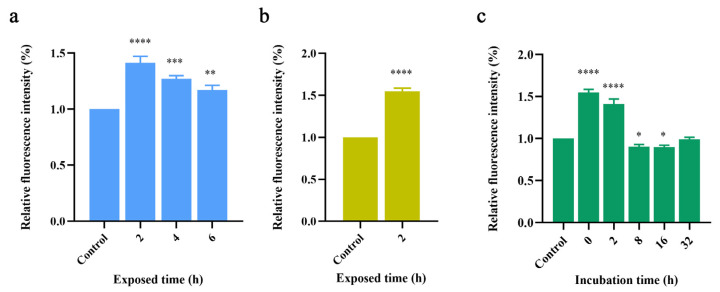
(**a**) Relative fluorescence intensities of experimental groups exposed to different irradiation times. (**b**) Relative fluorescence values of the experimental monomeric exposed group expressed as relative intensities. (**c**) Comparison of the relative fluorescence readings for the radiation-exposed groups at various incubation times. *: *p* < 0.05, **: *p* < 0.01, ***: *p* < 0.001, ****: *p* < 0.0001.

**Figure 2 ijms-24-05039-f002:**
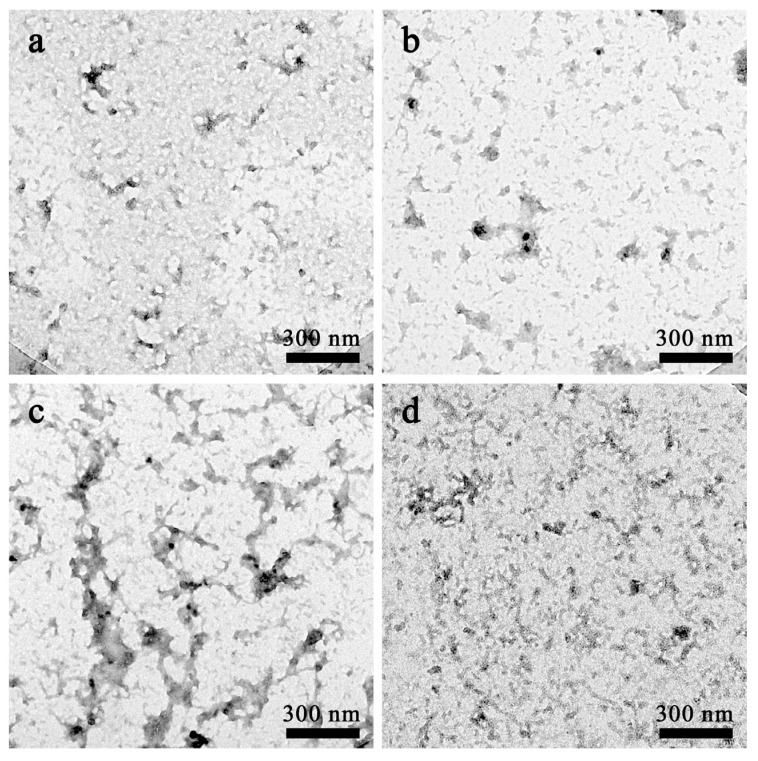
Aβ42 transmission electron microscope images. (**a**) Aβ42 monomer’s morphology after 2 hours of aggregation. (**b**) Aβ42 monomer’s morphology after 2 hours of aggregation under terahertz radiation. (**c**) Aβ42 oligomer’s morphology after 2 hours of aggregation. (**d**) Aβ42 oligomer’s morphology after 2 hours of aggregation under terahertz radiation. All scale lengths are 300 nm.

**Figure 3 ijms-24-05039-f003:**
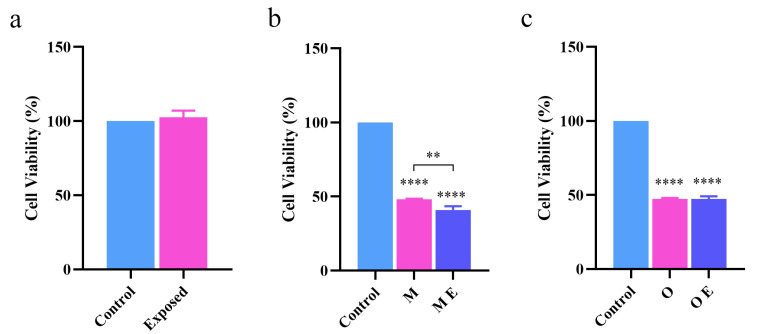
Cell viability under various interventional conditions. (**a**) The viability of PC12 cells exposed to terahertz radiation directly in the absence of Aβ42. (**b**) The viability of PC12 cells subjected to terahertz radiation in a mixed culture containing monomeric Aβ42. (**c**) The viability of PC12 cells that survived after being exposed to terahertz radiation in combination with oligomeric Aβ42. M, Aβ42 monomers; O, Aβ42 oligomers; M E, Monomer exposed to radiation; O E, Oligomer exposed to radiation. **: *p* < 0.01, ****: *p* < 0.0001.

**Figure 4 ijms-24-05039-f004:**
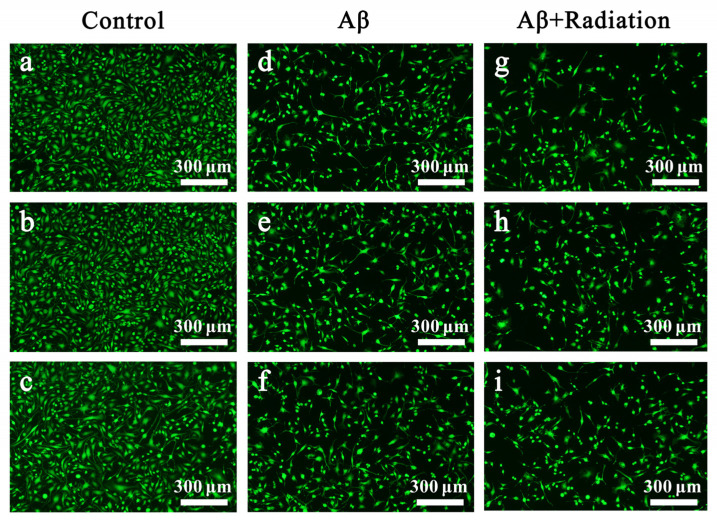
Live-cell staining under various interventional conditions. (**a**–**c**) Fluorescence images of PC12 cells without any intervention; (**d**–**f**) fluorescence images of PC12 cells after co-incubation with Aβ42; and (**g**–**i**) fluorescence images of PC12 cells following co-incubation with Aβ42 and terahertz radiation exposure.

**Figure 5 ijms-24-05039-f005:**
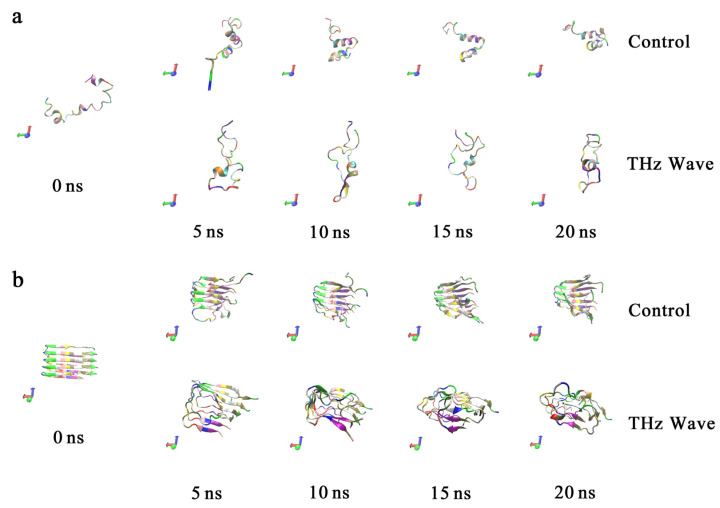
(**a**) Aβ42 monomer molecular dynamics trajectories. (**b**) Aβ42 oligomer molecular dynamics trajectories.

**Figure 6 ijms-24-05039-f006:**
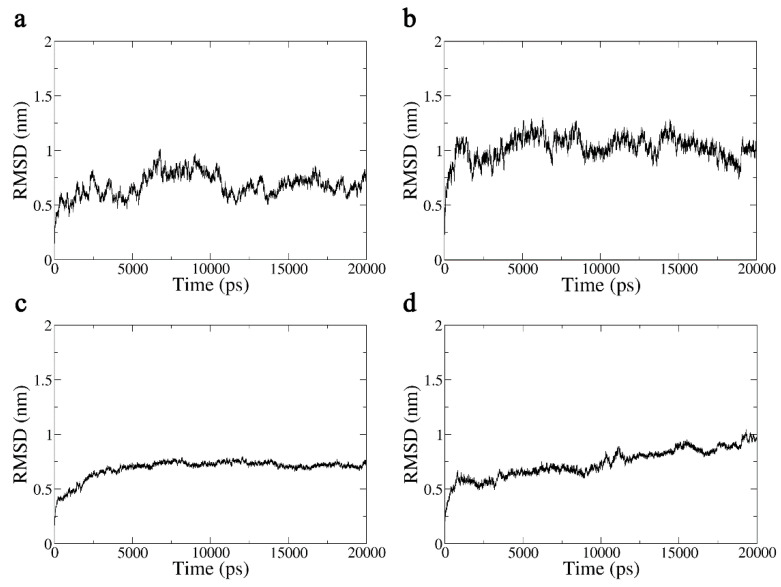
(**a**) Aβ42 monomer RMSD in 20 ns without an applied electric field. (**b**) Aβ42 monomer RMSD in 20 ns in a terahertz electric field. (**c**) RMSD of the hydrophobic fragment of the Aβ42 oligomer in 20 ns without an applied electric field. (**d**) Hydrophobic Aβ42 oligomer fragment RMSD in 20 ns in a terahertz electric field.

**Figure 7 ijms-24-05039-f007:**
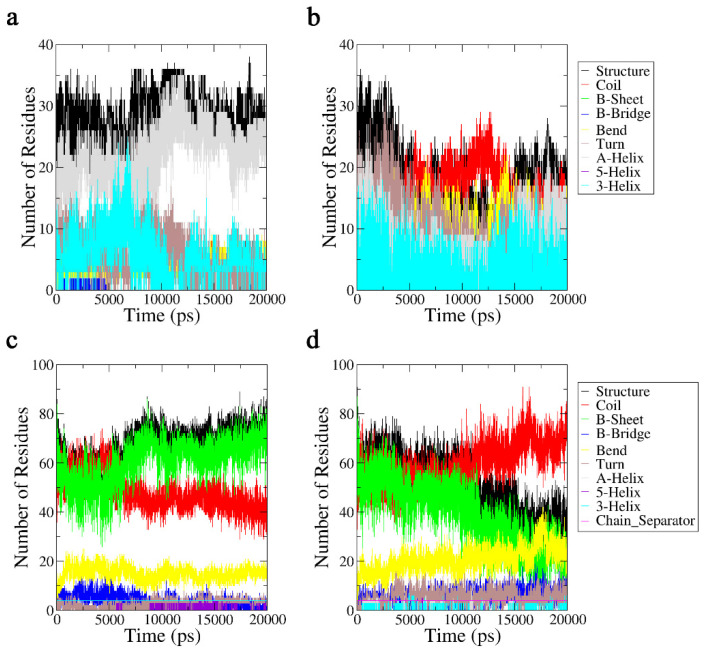
(**a**) Aβ42 monomer secondary structures in 20 ns without an applied electric field. (**b**) Aβ42 monomer secondary structures in a terahertz electric field in 20 ns. (**c**) Hydrophobic fragment of Aβ42 oligomer secondary structures in 20 ns without an applied electric field. (**d**) A hydrophobic fragment of the Aβ42 oligomer’s secondary structures in a terahertz electric field in 20 ns. Structure = α-Helix + β-Sheet + β-Bridge + Turn.

**Figure 8 ijms-24-05039-f008:**
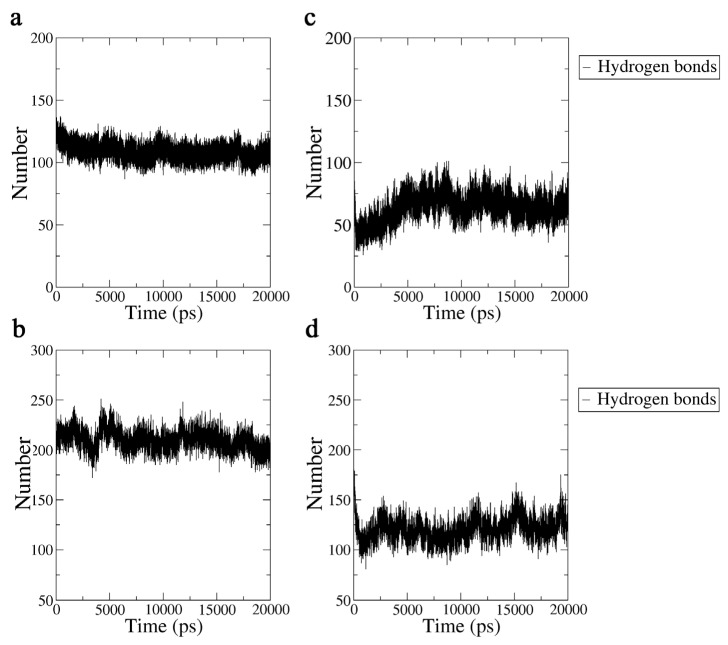
(**a**) The number of hydrogen bonds formed in 20 ns between the Aβ42 monomer and water molecules in the absence of an electric field. (**b**) The number of hydrogen bonds formed in 20 ns between the Aβ42 monomer and water molecules under a terahertz electric field. (**c**) The number of hydrogen bonds formed in 20 ns between the Aβ42 oligomer and water molecules in the absence of an electric field. (**d**) The number of hydrogen bonds formed in 20 ns between the Aβ42 oligomer and water molecules under a terahertz electric field.

**Figure 9 ijms-24-05039-f009:**
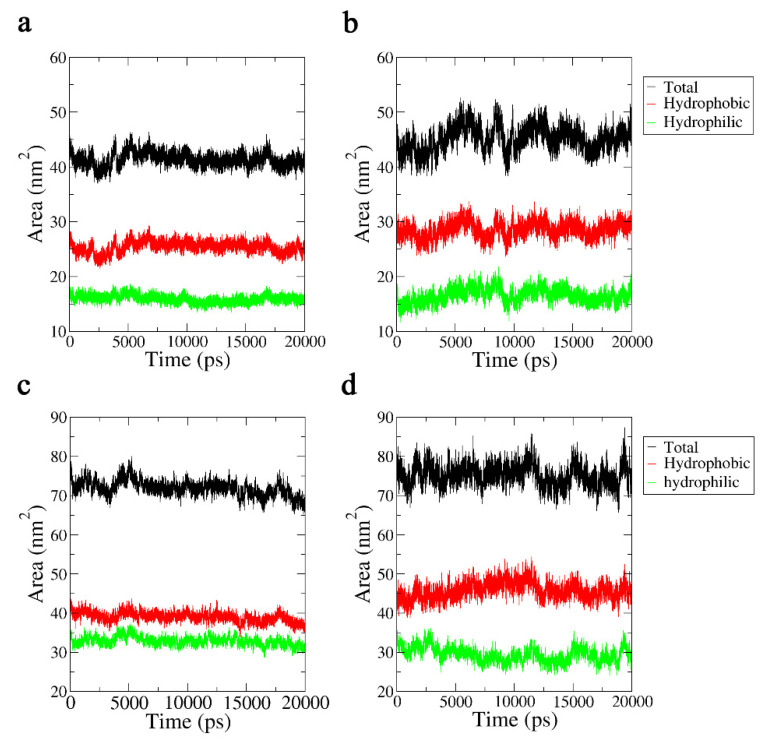
(**a**) SASA of the Aβ42 monomer in 20 ns. (**b**) SASA of the Aβ42 monomer in 20 ns under a terahertz electric field. (**c**) SASA of the Aβ42 oligomer in 20 ns. (**d**) SASA of the Aβ42 oligomer in 20 ns under a terahertz electric field.

**Figure 10 ijms-24-05039-f010:**
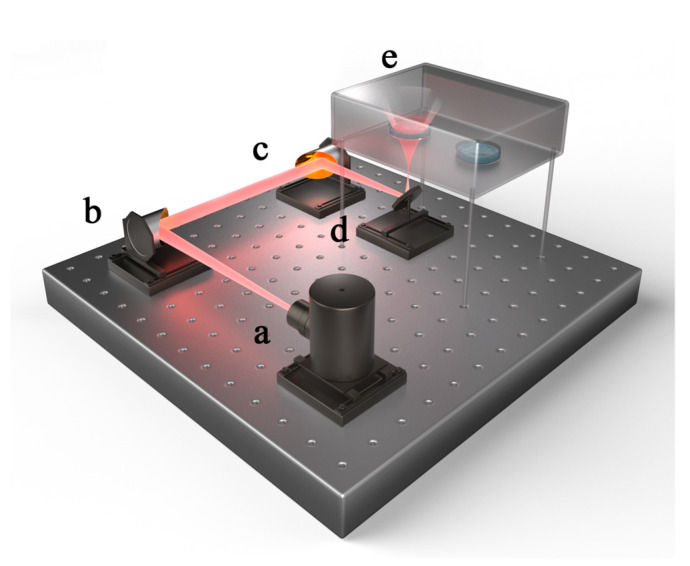
Schematic representation of the irradiation system. (**a**) a terahertz waves generator; (**b**) OAP 1; (**c**) OAP 2; (**d**) a plane mirror; and (**e**) an opaque chamber.

**Figure 11 ijms-24-05039-f011:**
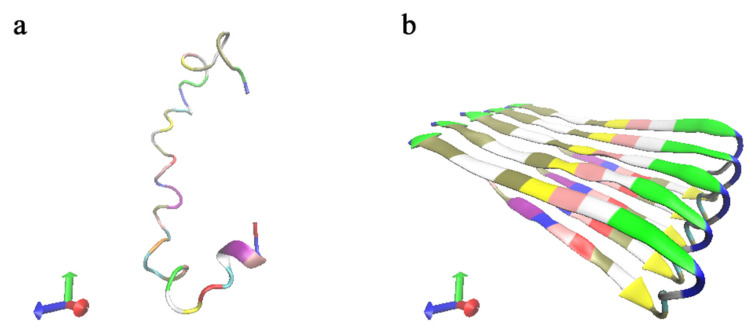
(**a**) The initial structure of the Aβ42 monomer simulation. (**b**) The initial structure of the Aβ42 oligomer simulation. The different colors represent the various amino acids.

**Table 1 ijms-24-05039-t001:** Statistics on the percentage of secondary structure and the average amount of hydrogen bonds.

	Secondary Structure (%)			Hydrogen Bonds (Number)
	Structure	α-Helix	β-Sheet	
Monomer	59	50	0	99
	57	32	0	100
	57	48	0	99
RSD	0.94	8.06	0	0.47
Monomer and Field	31	13	2	71
	18	2	2	64
	24	10	0	77
RSD	5.31	4.64	0.94	5.31
Oligomer	50	0	46	208
	47	0	44	201
	43	0	38	206
RSD	2.87	0	3.40	2.94
Oligomer and Field	37	0	30	125
	40	0	32	130
	36	0	29	140
RSD	1.70	0	1.25	6.24

## Data Availability

Not applicable.

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
