# Peer review of "Effects of Terahertz Radiation on the Aggregation of Alzheimer’s Aβ42 Peptide"

_ijms, 2023, doi:10.3390/ijms24055039_

Round 1

Reviewer 1 Report

The authors aim to study the effects of terahertz radiation on the aggregation process of Aβ peptide, the most well-known amyloidogenic protein since its amyloid fibrils are found in the brains of patients suffering from Alzheimer’s disease.  To achieve this, the authors utilized Thioflavin T fluorescence assay, Transmission Electron Microscopy (TEM), cell culture and toxicity assay, live cell fluorescence experiments as well as Molecular Dynamic simulations.  According to the results of this manuscript, terahertz radiation affects Aβ42 peptide amyloidogenesis.  More specifically, when Aβ42 amyloidogenesis is at the nucleation phase, radiation has an induction effect.  On the other hand, at the stage that protofibrils are starting to form, radiation has an inhibitory effect.

Although, this approach would be quite interesting for anyone researching for Alzheimer’s disease, there are a few shortcomings that should be addressed.

Major revisions:

11. The Materials and methods are not detailed enough and there are many ambiguities.

a.      The authors should use only the correct way to refer to Thioflavin T. Not only in the Materials and methods section, but also at the Results and Discussion section, they use either Thiosemicarbazone T or Thioflavin T.  According to literature, Thiosemicarbazones are ligands with ability to form stable complexes with Re(I) and Tc(I) that might be relevant to design potential radiopharmaceuticals with affinity for certain biomolecules. Additionally, there are studies for the design of thiosemicarbazone ligand with affinity for amyloid fibrils.  Thus, its of essential to use the right term when describing experimental procedures and results.

b.      What is the pH and the concentration of the PBS buffer?

c.       The preparation of TEM grids should be described in more detail.  How was the excess stain of 1% phosphotungstic acid removed? Also, the correct way to refer to TEM grids is “200-mesh ultra-thin carbon film grids” and not “200-purpose ultra-thin carbon film mesh”.

d.      In the Molecular Dynamics Simulations section, there is no mention at all how the effect of terahertz radiation is incorporated for conducting the MD simulations.  Also, the authors must clarify why they utilize different force fields for the monomer and the oligomer of Aβ42. Finally, the simulation time of 20 ns might not be adequate for drawing reliable conclusions and must be extended, if possible.

   2. In Results and Discussion section:

a.      Is the reference to Figure 3(b) in line 197 correct?

b.      According to ThT results, terahertz radiation’s inhibitory effect can be observed at the stage of aggregation of the oligomer to protofibril.  This is the exponential phase of amyloidogenesis and not its late stage as described in the abstract.

c.       The Transmission electron micrographs are of different magnification, and thus their scale bars should be different.  Also, the authors mention that for all TEM images, the scale bars are all 200 nm.  However, there is only one scale bar at figure 4d, and according to its label, corresponds to 300 nm.  The authors should add scale bars to every TEM image.  Also, it is advised to mention the magnification of each image at the respective figure legend.

d.      How the authors are sure that what they observe is indeed Aβ42 oligomers and not artifacts? Did they conduct any additional experiment, such as western blot analysis?

e.      The authors should mention the color code they used to color the Aβ42 structure.

Minor revisions:

   1. My concern is also focused on the writing.  This paper would benefit from some language editing, and at the very least, some good proofreading.

For example:

a.      The term “in vitro” should be in italics.

b.      In lines 39 and 41, the full stop should added after et al and not after the reference.  E.g. Zhang et al [16]. Should be Zhang et al. [16].

c.       Is the sentence in lines 48-50 correct? Which is the source of terahertz radiation?

   2. There are many grammatical errors. For example, in line 116, further is written with double f.  Also, in line 117, a blank is missing between “shaking” and “at”.

    3. The title is a little misleading.  Alzheimer’s disease is indeed associated with many proteins; however, this work is focusing only on Aβ42 peptide.

   4. The authors are advised to add why they chose to study the 42 residue-long form of Aβ peptide and not some other form.

Author Response

I appreciate your thorough review. I carefully read your feedback and respectfully replied. Please see the attachment for more information.

Reviewer 2 Report

Review on Effects of Terahertz Radiation on the Aggregation of Proteins Associated with Alzheimer's Disease

I have completed my review of manuscript ijms-2110069, entitled, Effects of Terahertz Radiation on the Aggregation of Proteins Associated with Alzheimer's Disease.”

Alzheimer's disease (AD) is a progressive neurodegenerative disorder that causes dementia and eventually death. There is currently no efficient treatment available to slow or stop the progression of AD. Terahertz waves are a discovery of a new type of non-ionizing radiation that is unique in that they may have an impact on the secondary bonding networks of biological systems, which may then have an impact on the course of biochemical reactions by changing the conformation of biological macromolecules. The findings of the proposed study showed that the 3.1 THz electromagnetic wave helps the Aβ42 monomer aggregate during the nucleation aggregation stage, but that as the aggregation degree increases, thus facilitating effect gradually decreases. The 3.1 THz electromagnetic wave, however, exhibits a suppressive effect during the final phase of fibrillogenic aggregation. The authors conclude that terahertz radiation affects the stability of Aβ42 secondary structure, which in turn affects how Aβ42 molecules are identified during the aggregation process and result in a seemingly abnormal biochemical response.

The subject and findings of this article are interesting and useful against AD. Before making a positive decision, I have some concerns and comments about the present form of the manuscript that must be addressed first.

Comments for authors

Comment 1: AD can be caused by a variety of factors. When comparing the number of factors that cause AD, the authors' description offered in the background (introduction) is insufficient to convey the information, which should be increased for new readers. The microwave was also thought to be responsible for AD. I encourage authors to add some background on this topic. The suggested article may assist authors in expanding their background knowledge and understanding the mechanisms by which the EM field interacts with and affects biological systems for various effects. The inclusion of this recent article could help to strengthen the introduction section.

Article: Microwave Radiation and the Brain: Mechanisms, Current Status, and Future Prospects. International Journal of Molecular Sciences vol. 23 (2022). [https://doi.org/10.3390/ijms23169288].

Comment 2: Section 2.2., “Radiation method” replace with “introduction of THz source and irradiation method.” Also, the irradiation method seems to be insufficient and needs to expand extensively. Is the temperature 37 ℃ is room temperature or temperature rise after exposure?

Comment 3: If Terahertz is already specified as short form "THz", no need to repeat its full form throughout the explanation. The subscript and superscript letters need to check carefully (e.g., line 124). Double-check the whole manuscript to remove such misleading errors.

Comment 4: In Figure 1, add legends. It is advisable to update Figure 1 to include the names of each component of the apparatus and details. Otherwise, it would just be a picture that readers couldn't possibly understand.

Comment 5: What is the difference between Figure 3 (a,b,c), for 2h results? It is advised to maintain the color consistency of the graphs in a single figure to avoid confusion? 6h and 8h appeared as green in (a) and (c). Also, in the figure 3 caption, various radiation, duration is might be not appropriate. 2h – 32h is the exposure time or incubation time?

Comment 6: Add a scale bar on each TEM photograph of Figure 4.

Comment 7: Figure 5, "Exposed" should be used in place of "Radiation." It does not represent radiation; rather, it compares the exposed group to the control group.

Comment 8: What are the findings in Figure 7? 0 ns – 20 ns is the simulation time? How did the authors conduct the simulation and take the THz exposure into account during the simulation? The details need to explain in the material and method section.

Comment 9: The paper contains errors and typos that make it difficult to understand and distort its intended meaning. I encourage authors to reread carefully and fix any grammatical errors.

Author Response

(The authors gave the same response as above.)

Round 2

Reviewer 1 Report

The authors have satisfactorily addressed all my comments in the revised version of the manuscript.  In its present form, this manuscript is recommended for publication.

Reviewer 2 Report

I have completed this manuscript in its revised form. The authors' responses to my comments and queries are greatly appreciated. The revised version is now worthy and my opinion is to publish this manuscript in IJMS in its present form.
